# Properties and Degradation Performances of Biodegradable Poly(lactic acid)/Poly(3-hydroxybutyrate) Blends and Keratin Composites

**DOI:** 10.3390/polym13162693

**Published:** 2021-08-12

**Authors:** Martin Danko, Katarína Mosnáčková, Anna Vykydalová, Angela Kleinová, Andrea Puškárová, Domenico Pangallo, Marek Bujdoš, Jaroslav Mosnáček

**Affiliations:** 1Polymer Institute, Slovak Academy of Sciences, Dúbravská cesta 9, 845 41 Bratislava, Slovakia; katarina.mosnackova@savba.sk (K.M.); anna.vykydalova@savba.sk (A.V.); angela.kleinova@savba.sk (A.K.); jaroslav.mosnacek@savba.sk (J.M.); 2Institute of Molecular Biology, Slovak Academy of Sciences, Dúbravská cesta 21, 845 51 Bratislava, Slovakia; andrea.puskarova@savba.sk (A.P.); domenico.pangallo@savba.sk (D.P.); 3Faculty of Natural Sciences, Institute of Laboratory Research on Geomaterials, Comenius University in Bratislava, Mlynská dolina, 842 15 Bratislava, Slovakia; marek.bujdos@uniba.sk; 4Centre for Advanced Materials Application, Slovak Academy of Sciences, Dúbravská cesta 9, 845 11 Bratislava, Slovakia

**Keywords:** biodegradable composites, keratin, polylactide, polyhydroxybutyrate

## Abstract

From environmental aspects, the recovery of keratin waste is one of the important needs and therefore also one of the current topics of many research groups. Here, the keratin hydrolysate after basic hydrolysis was used as a filler in plasticized polylactic acid/poly(3-hydroxybutyrate) blend under loading in the range of 1–20 wt%. The composites were characterized by infrared spectroscopy, and the effect of keratin on changes in molar masses of matrices during processing was investigated using gel permeation chromatography (GPC). Thermal properties of the composites were investigated using thermogravimetric analysis (TGA) and differential scanning calorimetry (DSC). The effect of keratin loading on the mechanical properties of composite was investigated by tensile test and dynamic mechanical thermal analysis. Hydrolytic degradation of matrices and composites was investigated by the determination of extractable product amounts, GPC, DSC and NMR. Finally, microbial growth and degradation were investigated. It was found that incorporation of keratin in plasticized PLA/PHB blend provides material with good thermal and mechanical properties and improved degradation under common environmental conditions, indicating its possible application in agriculture and/or packaging.

## 1. Introduction

One of the most important environmental challenges is pollution by nondegradable plastics used as one-way material mainly from the area of packaging materials. The problem becomes even more pronounced with partial disintegration of these plastics by UV radiation degradation, mechanical abrasion and biological degradation, which lead to the production of microplastics, which are harmful primarily due to their bioaccumulation and indirectly due to the toxic additives and microorganisms, adsorbed on the microplastics’ large surface areas, that enter the food web and, consequently, human food [1]. Therefore, there is growing interest in materials from renewable resources also with the vision of their biodegradability and recyclability. Among them, aliphatic polyesters represented mainly by polylactic acid (PLA) and polyhydroxyalkanoates (PHA), already commercialized on a large scale, are the most attractive. They possess mainly the ability to undergo both hydrolytic degradation and biodegradation by soil microorganisms in compost [1], while their limited mechanical properties and processability can be improved by blending and/or additive addition. Due to a semicrystalline structure, the PLA and poly(3-hydroxybutyrate) (PHB) have reduced flexibility and elongation at break. Therefore, plasticizers such as glycerol, triacetin, citrate esters or oils are commonly used to significantly reduce their stiffness and brittleness [2,3,4,5,6]. Their properties can be further improved and tuned by the addition of fillers such as clay [7,8], silica [9,10,11] or carbon black [12] or by copolymer compatibilization of polymer blends [13,14,15,16], while even synergistic effect in the improvement of the final blend or composite properties can be achieved [17,18]. Therefore, in addition to the emphasis on the research and development of polymer matrices, attention is also focused on the fillers. Keratin is an example of natural material produced in meat and leather processing as a side product contributing to the overall natural waste. Thus, an effort to investigate its further processing and use is highly demanded. In recent years, keratin-based products mixed with a variety of polymer matrices have already led to the development of materials for biomedical applications [19,20] taking advantage of some key properties of keratins [21]. Recent examples of keratin composites suitable for medical application showed enhanced cell proliferation on simple composite disks [22,23] or on electrospun nanofibrous mats for wound healing application [24,25]. The presence of keratin in PLA/hydroxyapatite fibrous membrane significantly enhanced bone formation as a result of the strong interactions between keratin and Ca^2+^ cations of hydroxyapatite [26]. Low keratin content (0.1–10 wt%) usually had a positive effect on the thermal and mechanical properties of final composites [27]. Recently, electrospun fibers made of 30 wt% high-molecular-weight keratin in poly(3-hydroxybutyric acid-*co*-3-hydroxyvaleric acid) nanofibrous mats demonstrated improved mechanical properties [28]. On the other hand, high loading content (up to 60 wt%) of chicken feather fibers in PLA and PBAT matrices led to lightweight and thermal insulating material, while its thermal stability, tensile strength and elongation at break were negatively affected [29]. Carrillo et al. [30] described similar decreases in PLA tensile strength and elongation at break of about 58% and 12%, respectively, at high (25 wt%) keratin content. Basic hair keratin is an insoluble and chemically resistant cysteine-rich mix of proteins forming nonhelix domains crosslinked by intramolecular disulfide bonds [31]. For its further processing and obtaining soluble products, a cleaving of crosslinks and/or carrying out hydrolysis of peptide bonds is required. The most commonly used technologies include hydrolysis by alkaline solutions, acids, reduction and oxidization agents or enzymes to obtain keratin hydrolysates suitable for further processing [32].

Recently we have shown successful blending of PLA and PHB semicrystalline polyesters compatibilized by acetyl tributyl citrate (ATBC) as plasticizers in composition PLA/PHB/ATBC 85/15/15 with significantly improved elongation at break [2,33,34,35]. This polymer blend was filled with carbon black [33,34] and tested as mulching foils. Polyester filled by acid hydrolysis treated keratin with filler content from 1 to 20 wt% was also prepared and characterized by mechanical and rheological analysis [2]. A relatively high content of functional groups of keratin hydrolysate, i.e., up to 10 wt%, did not lead to significant degradation of polyesters during the preparation by mixing processing at high temperature. Only at the highest investigated keratin hydrolysate content of 20 wt% was a more pronounced decrease in the original average weight molar mass of matrix polymer observed. The retention of good mechanical properties of high-molar-mass polyester matrices with appropriate plasticizing effect transferred into 140% of elongation at break for keratin loading of 20 wt% allowed preparation of composites exhibiting good mechanical properties for a wide range of applications.

In this work, PLA/PHB/keratin plasticized composites were prepared, while the keratin hydrolysate was obtained by alkaline hydrolysis providing mainly primary amine functional groups. In addition to the investigation of the thermal and mechanical properties, the hydrolytic ability of the composites at room temperature and microbial growth and degradation were investigated as well.

## 2. Materials and Methods

*Materials*. Poly(lactic acid) pellets PLA 4042 D (PLA, Nature Works, density 1.25 g cm^−3^) were obtained from Resinex Slovakia. Poly(3-hydroxybutyrate) powder (PHB, density 1.20 g cm^−3^) was obtained from Biomer (Krailling, Germany). The acetyl tributyl citrate (ATBC, Merck, 98%) and L-lysine (Merck, 98%) were used as received. The hydrolyzed keratin in the form of powder with particle size of 10–200 µm, supplied by VIPO, a.s. (Partizánske, Slovakia), was prepared by alkaline hydrolysis procedure from sheep wool described previously [36]. The average number molar mass of this product obtained based on PMMA standards calibration was *M*_n_ ≈ 3500 g mol^−1^. The concentration of primary amino groups in keratin was established by colorimetric reaction with 2,4,6-trinitrobenzene sulfonic acid (Merck, 5% (*w*/*v*) in H_2_O) according to standard procedure [37]. Extrapolated concentration 0.029 mmol g^−1^ of primary amino groups was obtained from UV-Vis calibration curve constructed for L-lysine in dimethylformamide at 420 nm.

The PLA/PHB 85:15 blend, PLA/PHB/ATBC 85:15:15 blend and PLA/PHB/ATBC/K-xy composites, where xy denotes 1, 3, 5, 10 and 20 wt% keratin were prepared by melt mixing in a Brabender Plasti-Corder (Duisburg, Germany) at 175 °C at 40 rpm (rounds per minute) for 10 min. Polymer foils were prepared either by solution casting or by compression molding: (i) Foils were prepared by casting 1 mL of polymer chloroform solution (5 g/100 mL) on a glass plate (28 mm × 35 mm). The solvent was slowly evaporated. The thickness of the foils was 50 µm. (ii) Foils were compression molded at 180 °C in two steps, first 1 min without pressure and additional 1 min under pressure at 2.65 MPa. Subsequently, the prepared foils were cooled down. The thickness of the foils was ~100 µm. For hydrolysis experiments, square-shaped samples with 15 mm sides were cut. The specimens with rectangular shapes of “dog bone” with dimensions of 75 mm × 4 mm (×12.5 mm in head) for mechanical testing and strips with dimension 30 mm × 5 mm for dynamical mechanical analysis were cut and stored before the measurements for approximately 24 h at ambient conditions. Polymer samples were further processed by annealing at 80 °C for 16 h to reset the thermal history of differently prepared samples. Samples prepared in these ways were used for thermal, mechanical and hydrolysis studies.

*Methods*. NMR spectra were recorded using a 300 MHz Varian spectrometer and deuterated chloroform solvent as a reference. Chemical shifts are expressed in parts per million (ppm).

The tensile tests were performed at room temperature using a Dynamometer Instron 4301 (Instron Corporation, Norwood, MA, USA) in accordance with standard ASTM D638. A testing rate of 1 mm/min was applied until 1% deformation was reached, and then the rates were increased to 50 mm/min. Average values of the tensile strength (*σ*_TS_), elongation at break (*ε*_B_) and Young’s modulus (*E*) were determined from the stress–strain curves.

Dynamic mechanical thermal analysis (DMTA) was performed using the Dynamic Mechanical Analyzer DMA Q800 (TA Instruments, New Castle, DE, USA) within the temperature range from −20 to 160 °C with a heating rate of 3 °C per min. The measurements were carried out in tensile mode at 1 Hz frequency with deformation amplitude of 40 µm. The storage modulus (*E*′), loss modulus (*E*″) and loss tangent (tan *δ* = *E*′/*E*″) were determined for at least three specimens of each sample formulation.

DSC measurements of all samples were performed on DSC 821 Mettler Toledo (Mettler-Toledo, Greifensee, Switzerland) equipped with an intercooler in two parallels. Samples of 4–5 mg were placed into sealed aluminum pans with one pinhole on the lid. Nitrogen at 50 mL/min was used as a purge gas. Each sample was heated from 20 to 200 °C at 5 °C/min, held at this temperature for 2 min, cooled to 20 °C at 5 °C/min and again heated to 200 °C. The temperature scale was calibrated to the melting points of indium and zinc; the enthalpic scale was calibrated to the enthalpy of fusion of indium and zinc. The degree of crystallinity was calculated with Equation (1):(1)$$\chi_{c}=\left[\frac{\Delta H_{m}}{\left(1-\phi \right)\Delta H_{m}^{c}}\right]\cdot 100\%$$
where *ϕ* is the weight fraction of the dispersed phase in the blend; Δ*H*_m_ is the melting enthalpy; and $\Delta H_{m}^{c}$ is the heat of melting, with a value of 93.1 J g^−1^ for crystalline PLA and a value of 146 J g^−1^ for crystalline PHB, when the proportion of PLA and PHB in the blend is 85/15% [38].

Thermogravimetric analysis was performed in two parallel measurements for each sample. The measurement was carried out at a temperature range of 25–600 °C on a Mettler Toledo TGA/SDTA 851^e^ (Mettler-Toledo, Greifensee, Switzerland) instrument under a nitrogen atmosphere (80 mL/min) and a heating rate of 10 °C/min. Samples of 4–5 mg were placed in aluminum pans. Indium and aluminum were used to calibrate the temperature. The first derivative of TGA curves (DTG) was obtained by the analysis STAR^e^ evaluation software.

The morphology of films was inspected by optical microscopy using digital microscope Leica DVM6 M (Leica Microsystems, Heerbrugg, Switzerland) equipped with objective PlanAPO FOV 12.55 using 150× optical magnification.

Fourier transform infrared attenuated total reflectance spectroscopy (FTIR-ATR) or transmission NIR was applied for investigation of the chemical composition. The MIR-NIR measurements were performed using a Nicolet 8700^™^ spectrometer (ThermoScientific, Madison, WI, USA) with a resolution of 4 cm^−1^ and scan ranges of 4000–400 cm^−1^ and 10,000–4000 cm^−1^ for MIR and NIR, respectively. A Shimadzu UV-1650PC spectrometer (Shimadzu Corporation, Kyoto, Japan) was used for UV-Vis measurements.

The molar masses were estimated by gel permeation chromatography (GPC) using trifluoroethanol (TFE) as an eluent with addition of 0.1 M potassium trifluoroacetate to increase ionic strength for determination of polymer molar masses and in dimethylacetamide with addition of 0.1 wt% LiBr eluent for keratin molar mass determination. The GPC system consists of Shimadzu LC-20 pump (Shimadzu Corporation, Kyoto, Japan), Shimadzu refractive index detector (Shimadzu USA Manufacturer Inc., Canby, OR, USA) and two PPS PFG 5 µm columns or PSS GRAM 5 µm columns (*d* = 8 mm, *l* = 300 mm; 100 + 1000 Å) at 25 °C. Poly(methyl methacrylate) standards (Polymer Standard Services, Mainz, Germany) were used for calibration. For GPC, the polymer samples were dissolved in a mixture of 1,1,1,3,3,3-hexafluoro-isopropanol (HFIP) and TFE with *v/v* ratio 1/9. First, 2 mg of polymer sample was treated with 100 µL of HFIP to disrupt the crystalline structure of PHB and PLA, followed by addition of 900 µL of TFE.

*Hydrolysis experiments*. Hydrolysis of the films was performed in a 0.01 M NaOH solution (pH = 11.8) or in extra-pure water (MilliQ pH = 7) at 25 °C for predetermined periods of time. The NaOH solution was exchanged every 10 days. The content of extractable products, after intensive washing with distilled water at room temperature 3 times and drying under vacuum (M), was determined from the change in mass of the polymer films before (M_0_) and after hydrolysis, which can be written as follows:(2)$$\mathrm{Extractable}\;\mathrm{products}\;(\%)=\frac{M_{0}-M}{M_{0}}\times 100\%$$

After washing out the extractable products, the residual nonhydrolyzed parts of the films were analyzed by GPC.

*Microbiological analysis*. In order to assay the polymer biodegradation, a soil community was used. A portion of 10 g (wet weight) of the soil was mixed in a sterile 250 mL Erlenmeyer flask with 90 mL of a 0.9% (*w/v*) NaCl solution and incubated at 28 °C in an incubator by shaking at 90 rpm for 2 h. The suspension was filtered through Whatman 1 filter paper (Merck, Darmstadt, Germany), and this filtered soil suspension with pH~5.8 contained the microbial community used to assay the biodegradability of polymers. Polymer composite samples (squares, 10 mm × 10 mm) were placed in a 100 mL Erlenmeyer flask with 10 mL of soil suspension. The flasks were incubated at 28 °C for 4 weeks. After culturing, the polymer samples were separated from the soil suspension and were carefully rinsed twice with sterile phosphate-buffered saline (PBS) to remove planktonic cells and subsequently washed by shaking in sterile PBS for 10 min to remove adherent cells. The suspension with the cells from polymer surfaces was serially diluted (by a total factor of 108), and 100 µL of each dilution was inoculated to a Petri dish with nutrient agar. The plates were incubated at 28 °C for 24–48 h, and the concentrations of the bacterial colonies were calculated as colony-forming units (CFU)/mL.

## 3. Results and Discussion

### 3.1. Preparation and Chemical Characterization of Prepared Keratin Composites

The hydrolysis of raw keratin waste is an important process for the utilization of keratin in polymer composites. The effort is to break crosslinked structure of disulfide bridges of cysteine-rich keratin peptide and introduce functional groups for better miscibility with polymer matrix. Details of alkaline hydrolysis and characterization of used keratin hydrolysate are presented in [36] and Section 2, respectively.

Here we prepared five keratin composites with filler contents of 1, 3, 5, 10 and 20 wt% dispersed in PLA/PHB/ATBC matrix with the composition ratio of 85/15/15. Two polymer blend samples, PLA/PHB and PLA/PHB/ATBC, without keratin as model matrices were prepared by the same methodology. The presence of keratin in composites was confirmed by ATR-FTIR spectroscopy in the MIR region as well as by transmission mode in the NIR region. The main characteristic absorption bands of keratin composite are shown in Figure 1a. Strong carbonyl amide stretching vibration and -N-H bending vibration related to keratin protein were observed at 1650 and 1520 cm^−1^, respectively (Figure 1b). Keratin-related absorption bands are clearly distinguished from those attributed to carbonyl ester stretching of the polyester matrix at 1720 cm^−1^. Additionally, general -CO-NH- vibrations were visible in the NIR region at 4860 cm^−1^. These absorption bands were clearly visible for composites with at least 3 wt% keratin content and progressively increased with keratin loading (Figure 1c).

Polymer composites prepared by solution casting or compression molding with the thickness of 50 or 100 µm, respectively, were semitransparent films. Keratin particles caused slightly yellow coloring mainly for high keratin content. Optical microscopy pictures confirmed the presence of keratin particles with a relatively broad size distribution from 10 to 200 µm in diameter (Figure 1d). The number of particles progressively increased with keratin content. These particles were well dispersed in the matrix for composites with keratin content of up to 5 wt%, while areas with aggregates appeared for higher loading ratios (Appendix A). The surface of keratin particles appeared as wet, indicating interactions between the keratin protein and polyester matrix in the matrix/filler interface (Figure 1d).

Molecular characteristics of polyester polymers after melt blending with keratin were evaluated by GPC analysis. PLA and PHB, like many aliphatic polyesters, can degrade upon melt blending; thus, processing affects the thermomechanical features of the blended material [39]. Significant degradation of these polyesters during processing at temperatures over the melting can affect the final properties of materials and consequently their application. Additionally, it was reported that the addition of low-molecular-weight additives, such as the commonly used plasticizer glycerol, can extensively increase the process of degradation [40]. GPC traces of neat PLA/PHB/ATBC showed a peak with *M*_p_ of ~45,000 g mol^–1^ related to the main PLA component accompanied with shoulder toward lower elution volumes (higher molar masses) related to PHB, i.e., approximately at the same elution volumes as neat PLA and PHB components before blending (Figure 2a), indicating no or negligible degradation during 10 min of melt blending at 175 °C. On the contrary, the addition of keratin with surface-rich functional groups such as primary amino groups during melt blending led to a progressive decrease in both *M*_n_ and *M*_w_. This is one of the differences of these keratin composites compared to similar composites prepared from keratin processed by acid hydrolysis [2]. The evolution of both molar masses and dispersity (*Ð*) with increasing keratin content is shown in Figure 2b (red circles) and listed in Appendix A. The most significant impact of processing on molecular characteristics was observed for 20 wt% keratin content with a decrease in molar masses by ~30%.

Two sets of polymer foils were compared during processing. Solution casting methodology for foil preparation did not require additional thermal treatment; on the other hand, a compression molding method can be more useful from the practical and industrial point of view. As confirmed by GPC measurements (Appendix A and Figure 2b, black squares), additional processing at 180 °C for 2 min during molding further affected the molecular characteristics of the polymer matrices. A slight decrease in *M*_n_ accompanied by an increase in dispersity indicated possible transesterification reactions. It is pertinent to note that with such a limited extent of *M*_n_ decreasing, no difference in mechanical properties of tested foils was observed, as is discussed below.

### 3.2. Thermal Properties of Keratin Composites Inspected by DSC and TGA Analyses

Thermal analyses characterize polyester materials from the point of view of the crystallinity degree and thermal stability. Both parameters are important for further applications and degradation (hydrolysis). Generally, the first heating run provides information about the physical structure of the material after certain physical aging affected by the aging history of the sample. In order to equilibrate the samples and promote the physical aging of the samples, an annealing at 80 °C for 16 h was applied prior to analysis. Annealing at the temperature over the glass transition temperature (*T*_g_) of the plasticized PLA/PHB blend allowed polymer segments to move and accelerated the secondary crystallization. The first heating run (Figure 3a) therefore showed only less pronounced glass transition and the set of melting endotherms related to PLA and PHB. No secondary crystallization was observed during the heating run.

Parameters such as *T*_g_, the secondary crystallization temperature (*T*_cc_) and the melting temperature (*T*_m_), collected in Table 1, were obtained from the second heating run of PLA/PHB and PLA/PHB/ATBC blends and the keratin composites with varying keratin content from 1 to 20 wt%. DSC traces of the second heating run are described in Figure 3b. Nonplasticized PLA/PHB blends exhibit glass transition temperature at 54.1 °C and melting endotherm at 152.6 °C corresponding to PLA fraction, while for PHB, melting at 173.3 °C was observed [17,41]. A wide exotherm centered at 125 °C can be attributed to secondary crystallization [42]. PLA and PHB components are considered immiscible, but a detailed study suggested their molecular interaction [17]. Significant differences were obtained upon addition of ATBC plasticizer. A decrease in *T*_g_ by 19 °C and a decrease in melting by 5 to 10 °C were observed for both components. The crystallization peak was also shifted to a lower temperature and became narrower. Moreover, four peaks of melting temperature appeared in the plasticized sample. It can be assumed that the first and second peaks at 135 and 147 °C correspond to the melting of PLA and the third and fourth peaks at 157 and 163 °C correspond to the melting of PHB. It was shown before that existence of two types of crystals, those formed during cooling and those pure crystals formed in second crystallization during heating, can be observed for PLA/PHB blends [17,43]. For both polymers, “as-formed” crystals of the second component in the immiscible blend can act as nucleating agents for the formation of larger crystals.

The addition of keratin in composites did not significantly affect the thermal behavior of the plasticized PLA/PHB/ATBC matrix. Keratin hydrolysate is an amorphous material, which does not exhibit its own glass transition and melting. The small plasticization effect of keratin can be attributed to the lowering of *T*_g_ with increasing keratin content from 35 °C for neat PLA/PHB/ATBC to ~30 °C for 20 wt% keratin (Table 1). On the other hand, the main melting peaks of PLA and PHB remain almost unaffected upon the addition of keratin.

The degree of crystallinity evaluated from first heating runs was the highest for neat matrices, approximately 47%, and varied between 44 and 32% for composites. A slight decreasing trend in *χ*_c_ with the addition of keratin could be observed. Similarly, the enthalpies of melting Δ*H*_m_ decreased. Interaction of the keratin particle surface with the polymer chains can decrease chain mobility and suppress the crystallization process. This is in an agreement with optical microscopy inspection, where the wet surface of the keratin particles in the composite was observed, indicating polymer–particle interactions.

The above-described results were obtained from polymer foils prepared by the solution casting method. For a comparison, polymer foils prepared by compression molding were investigated as well, since additional thermal treatment during compression at 180 °C for 2 min could affect the degradation degree of polyesters, as observed from GPC results, and thus can change their properties. Obtained thermal parameters and DSC records of all hot-pressed samples are shown in Appendix A and Appendix A. Slightly lower melting temperatures of PLA and PHB components in neat PLA/PHB blend were recorded. The keratin composites exhibited rather comparable parameters independently of sample preparation procedure. Here we can conclude that short high-temperature treatment of hot-pressed samples did not significantly affect the overall thermal properties of PLA/PHB/ATBC/keratin composites.

Thermogravimetric analysis showed that contrary to bare PLA, PHB and keratin materials, degrading in one significant step (Appendix A), the PLA/PHB blend degraded in two steps, and the plasticized PLA/PHB/ATBC matrix with all keratin composites exhibited the mass loss in three steps (Figure 4). First mass loss started at ~170 °C, where evaporation of ATBC plasticizer with the known boiling point at 172 °C can be expected [3]. The mass loss of material in the first step is around 10–13%, corresponding to ATBC content in the sample. The second and the third steps of thermal degradation can be assigned to the degradation of PHB and PLA, respectively. TGA courses of all plasticized samples, including those with keratin, were similar (Figure 4a). Particular peaks for all samples determined from DTG curves (Figure 4b) are summarized in Table 2. *T*_onset_ as the main thermal degradation started for the second step was the lowest for the nonplasticized PLA/PHB blend and continuously increased with plasticizing and keratin content. This was connected with the shift of the PHB degradation peak in DTG to the higher temperature. On the other hand, PLA degradation peak *T*_PLA_ was highest for K-1 and then decreased to be lowest for K-20 composite. It is also shown that the PHB peak begins to disappear at a higher dose of keratin. This could suggest that keratin interacts preferably with PHB and partially stabilizes it, as the *T*_PHB_ for the bare polymer is lower than those for all keratin composites. The carbonized residue at 600 °C (*R*_600°C_) was visible for the sample with 3 wt% keratin and reached ~5% for the K-20 sample.

### 3.3. Mechanical Properties

Mechanical properties of prepared polymer composite films are important characteristics for their practical applications and are strongly affected by both the addition of plasticizer and filler loading. Here, standard tensile tests were first performed, and stress–strain curves were evaluated, as shown in Figure 5. The unfilled PLA/PHB blend before plasticizing showed brittle behavior, as is clearly seen from the stress–strain curve in inset of Figure 5b. The addition of the plasticizer led to a significant increase in mobility of the polymer chains accompanied by one to two orders increased elongation at break as compared with nonplasticized PLA/PHB blend [2]. Generally, with an increased keratin loading, a decrease in both elongation at break and tensile strength was observed (Figure 5a). This reflects better formability and worse plasticity as a consequence of limited chain mobility in the amorphous region leading to insufficient options for chain straightening under tension. Stress–strain curves of neat matrix and keratin composites showed similar shapes without significant changes in the stress–strain slope, besides the sample with the highest keratin loading (Figure 5b). Initial maxima of tensile strength observed for composites disappeared for the sample with 20 wt% keratin content represented as yield strength due to the more elastic structure of materials. Common interaction of polymer chains present in amorphous phase and filler particles on the surface may prevent its further crystallization, i.e., hardening. It was reported [44] that higher keratin loading (greater than 10 wt%) leads to significant deterioration of tensile properties due to worse compatibility and disintegration into the material. Here, though values of elongation at break and tensile strength of keratin composites decreased similarly to those for neat PLA/PHB/ATBC plasticized blend (*ε*_B_ ≈ 300% and *σ*_M_ ≈ 24 MPa), polymer composites with high keratin content of 10 or 20 wt% still showed good mechanical properties, with values of elongation at break of 160 and 143% for 10 and 20 wt%, respectively.

Dynamic mechanical thermal analysis (DMTA) is a powerful technique for sensitive characterization of viscoelastic properties of polymers and composite materials. The total response extensively depends at the phase level or on molecular structure within a polymer, as well as multiphase composite systems. The behavior of loss factor (tan *δ*) and storage modulus (*E*′) for the nonplasticized PLA/PHB blend, the blend after the addition of ATBC and all prepared keratin composites are compared in Figure 6. The PLA/PHB blend without the addition of plasticizer exhibits two peaks ascribed to glass transition temperature (*T*_g_) of PHB and PLA due to the immiscible polymer system [18]. The addition of ATBC resulted in a typical plasticizing effect leading to significant shifting of *T*_g_ to lower value, while the shape/height remained unchanged.

The presence of the keratin slightly shifted the maximum of *T*_g_ to the lower temperatures, as was similarly observed in DSC analyses, and the height of loss factor maximum was also reduced. This is a consequence of the protein-like structure of keratin acting as a steric barrier for relaxation processes needed for the arrangement of polymer chains resulting in limited mobility. The increasing keratin loading led to a progressive decline in *T*_g_ as a result of higher free volume, which is usually observed as a result of plasticizing. In our previous study [2], we showed that the shifting of *T*_g_ to lower temperatures with increasing filler loading can be caused by transesterification during thermal processing leading to the shortening of polymer chains and thus higher chain mobility. The similarity of the obtained values suggests that keratin filler in this study affected the mechanical properties in a similar manner.

The temperature dependence of *E*′ is presented in Figure 6b. Generally, storage modulus is usually associated with the load-bearing capacity of the materials. The changes in *E*′ are frequently related to the interface adhesion between polymer phases and filler particles within composite materials and can be a consequence of degradation, crosslinking or crystallization. As is clearly shown in Figure 6b (black line), the *E*′ of nonplasticized and unfilled PLA/PHB blend showed a gradual decrease from −20 to 50 °C and then decreased rapidly because of the glass transition. The addition of ATBC resulted in shifting of glass transition and a sharp decrease in *E*′ at 25 °C as a result of material plasticizing and reduction in the strong interchain interactions, which promoted enhanced chain mobility resulting in higher flexibility of the PLA/PHB/ATBC blend.

For the keratin composites, there is an obvious increasing trend in *E*′ below *T*_g_ up to 5 wt% keratin loading due to stiffening of the material. The addition of the lowest content of keratin (1 wt%) into the PLA/PHB/ATBC blend led to an increase in *E*′ value at 25 °C, reflecting pronounced stiffness as a result of the reinforcing effect. Contrary to that, increasing keratin content above 10 wt% led to the dramatic decline in *E*′ at 25 °C, reaching up to a 30% decrease in *E*′ value compared to unfilled PLA/PHB/ATBC blend, indicating progressive softening of the composite.

The storage modulus in the glassy state is determined predominantly by the strength of the intermolecular forces and the way the polymer chains are aligned. In the case of composites, the storage modulus is additionally a function of filler stiffness and content [45]. Many studies showed that *E*′ modulus is not determined absolutely by the content of the filler. Ferreira et al. [46] attributed this behavior to the ability of graphene oxide used as filler to reduce the crystallinity of PLA. Wang et al. [47] reported that the reinforcing effect of organic–inorganic hybrid fillers at low quantities of filler was more effective than at higher content. In accordance with our results, the authors described that the composite with 20 wt% filler (aluminum hydroxide-based materials) had a lower *E*′ than the composite with 10 wt%, accompanied by lower stiffness.

### 3.4. Hydrolytic Degradation of Keratin Composites

The degradability of polyester/keratin composites, as materials with potential applicability in agriculture and/or packaging, under hydrolytic conditions is one of the most important characteristics. Hydrolytic degradation of the PLA/PHB/ATBC blend and its keratin composites at various pH values at 25 °C was investigated and followed by sample weight loss, DSC and GPC. It should be pointed out that, without considering additional effects of biodegradation and photodegradation, hydrolysis at pH~7 and at 25 °C can be taken as a model condition for open-field degradation of mulching foils. Although realistic hydrolysis at pH~7 is rather slow, it can be accelerated under basic or acidic conditions [48] to receive the results in a more reasonable time. It should be pointed out that hydrolysis of PLA in 0.1 M NaOH solution at pH~13.5 was described to be very fast, i.e., proceeded within 3 days [49]. In the case of PLA/PHB/keratin composites, it was already shown that molar masses decreased during processing. In addition, the polar character of keratin protein might also affect the rate of hydrolytic degradation of these composites. According to the discussion above, the hydrolysis in 0.01 M NaOH solution was performed first. An important factor is the thermal history of the samples because hydrolysis is faster in the amorphous phase and thus the degree of crystallinity affects this process. To minimize the effect of the sample history, the treatment of the samples at 80 °C for 16 h was carried out prior to hydrolysis.

As can be seen from Figure 7a, the progress of hydrolysis expressed as extractable products was faster for keratin-containing samples than for neat matrices and was proportional to the keratin content. While ~50% hydrolysis of neat nonplasticized and plasticized samples occurred after 50 days, all keratin-containing samples achieved this extent after ~20 days of hydrolysis under the basic conditions used. Faster hydrolysis at the beginning for PLA/PHB/ATBC sample could be attributed to the polar character of the keratin filler and partially also to slightly lower crystallinity proved by DSC and plasticizing effect. Fragmentation of foils occurred at ~50% extent of hydrolysis together with loss of transparency (Figure 7c). The final data obtained in the experiment were taken at 80% of sample weight loss. This point was achieved after 15 and 28 days for samples with 20 and 10 wt% keratin content, respectively, while for neat matrices, only ~50% of sample weight loss was achieved after a prolonged period of basic hydrolysis, i.e., up to 50 days. A similar course of hydrolysis with faster degradation for keratin samples was observed in pure water at pH~7. However, much slower overall hydrolysis with 10 to 30% of extractable products for PLA/PHB/ATBC and PLA/PHB/ATBC/K-10, respectively, was observed under these conditions (Figure 7b).

Table 3 presents the evolution of molar masses of neat matrices and their keratin composites before hydrolysis, after 14 days of hydrolysis and at the end of hydrolysis test, i.e., in the last points of the sample hydrolyses in Figure 7a. After 14 days of hydrolysis, the decrease in molar masses was only up to 10% for all tested samples. A certain polymer chain scission can be expected based on the broader molar mass distribution for particular samples. Such a low extent of degradation suggests surface erosion degradation with fully soluble extractable products rather than bulk matrix degradation. A different situation with bulk matrix degradation for PLA/PHB/ATBC blend and keratin composites was previously suggested during hydrolysis at an elevated temperature above the *T*_g_ of polyesters [2,50]. In that case, a consecutive degradation with time of hydrolysis with a decrease in molar masses down to 50% and 20% of original values was observed after 8 and 18 days, respectively [2].

More pronounced degradation with lower values of average *M*_n_ and *M*_w_ and bimodal distribution with appropriate wide *Ð* were observed in GPC traces at the end of the hydrolysis experiment. However, the presence of individual peaks from both PLA and PHB polymers at the end of hydrolysis was visible in GPC traces. In the case of the PLA/PHB/ATBC blend and the low-keratin-content composite PLA/PBH/ATBC/K-1, the slight shift of PLA peak toward lower molar masses was accompanied by the appearance of a high-molar-mass shoulder attributed to PHB (Figure 8a,b). On contrary, different elugrams with distinguished bimodal molar mass distribution were observed for composites with over 5 wt% keratin content (Figure 8c,d). The high-molar-mass component related to PHB is much more pronounced, and the PLA signal is significantly shifted toward the higher elution volume, i.e., toward lower molar masses. As can be revealed from ^1^H NMR spectra (Appendix A), the PLA/PHB ratio obtained from integration of methyl (-CH_3_) doublet proton signals of PLA at 1.56 ppm and PHB at 1.26 ppm decreased after the hydrolysis, while the decrease was more pronounced for samples with higher keratin content, i.e., with higher extent of hydrolytic degradation (Table 3).

DSC thermograms of samples after hydrolysis were evaluated in order to explain the mechanism of hydrolysis. Data obtained from DSC are summarized in Appendix A and Appendix A. Overall crystallinity degree slightly increased during hydrolysis from 47 to 49% for the neat plasticized blend. The increase in *χ*_c_ at approximately 5–15% was more pronounced for composites with 5–20 wt% keratin content. This indicates a higher extent of hydrolysis in the amorphous phase. A higher ratio of amorphous phase in composites with higher keratin content thus led to more pronounced hydrolysis. Moreover, the plasticizing effect of ATBC plasticizer was lost during hydrolysis since the glass transition temperature increased from 30–35 °C before hydrolysis to 50–55 °C at the end of hydrolysis.

### 3.5. Microbial Test

Although PLA undergoes mainly hydrolytic chain scission in the natural environment [51], PHB is known to undergo a bacteria-catalyzed biodegradation process [52]. On the other hand, crosslinked structure of keratin protein may require a special type of microorganism for the accommodation and degradation of the keratin source [53]. Here, a simple soil suspension containing various microorganisms was applied to assay the biodegradability of the polymer/keratin samples. For all tested samples, a 2–3 orders increase in the number of adherent microorganisms was observed compared to soil suspension without polymer (Table 4). The number of grown microorganisms slightly decreased with increased keratin content. Even though this could indicate that specific keratin protein in the composite inhibits the growth of microorganisms, the decrease was not noticeable. GPC analysis of foils after the 1-month experiment showed a slight decrease in average molar masses compared to those before the microbial experiment (Appendix A). Considering the results from the hydrolytic degradation experiments, it can be expected the observed decrease in molar masses in this experiment can correspond to hydrolysis rather than accelerated microbial degradation. On the other hand, partial participation of microorganisms in the degradation cannot be ruled out, and more intensive study of microbial and hydrolytic degradation will be needed in order to understand the contribution of each. Despite this, we can conclude that polymer foils made of PLA and PHB polyesters and plasticized by ATBC filled by keratin did not exhibit antimicrobial properties; microorganisms can freely adhere to the foil surfaces and thus could contribute to the biodegradation processes of these materials.

## 4. Conclusions

Plasticized PLA/PHB/ATBC/keratin composites in the form of polymer foils with various keratin contents in the range of 1 to 20 wt% were prepared. Up to the keratin content of 5 wt%, the keratin particles showed good dispersion within the matrix with the indication of surface particle/polymer interaction. Keratin partially promoted the degradation of matrices during the melt mixing, as confirmed by the decrease in the molar mass of the polymer matrix. The addition of keratin did not significantly affect the thermal behavior of the plasticized PLA/PHB/ATBC blend, while a negligible additional plasticization effect was visible in the slight *T*_g_ decrease after keratin addition. Thermal stability expressed by *T*_onset_ reached over 280 °C for all samples. Even though the mechanical properties of plasticized PLA/PHB/ATBC blend were negatively affected by keratin addition, even at the highest keratin loading of 20 wt%, the toughness and the flexibility were still sufficiently high for a wide range of applications. The matrices and keratin composites were subjected to hydrolytic degradation in ambient conditions, and the rate of degradation increased with increasing keratin content. Polar functional groups of keratin hydrolysate resulting from basic hydrolysis could be responsible for higher water molecule access and together with lower overall crystallinity of keratin composites facilitated hydrolytic degradation. Degradation proceeds preferably in the amorphous phase by erosion mechanism, especially in the first stage of degradation. In the later stage of hydrolysis, significant shifting of the PLA peak toward lower molar masses in GPC traces indicated a change of the mechanism to bulk degradation of PLA. Although both polymers were subject to hydrolysis, decreased PLA/PHB ratio was observed at the end of hydrolysis. The hydrolytically more stable polymer crystals of PHB were probably responsible for the slower degradation of this fraction. A microbial test confirmed free accommodation and proliferation of adhered microorganisms on the keratin composites, thus enabling their partial contribution to the (bio)degradation processes of the materials in the environment.

## Figures and Tables

**Figure 1 polymers-13-02693-f001:**
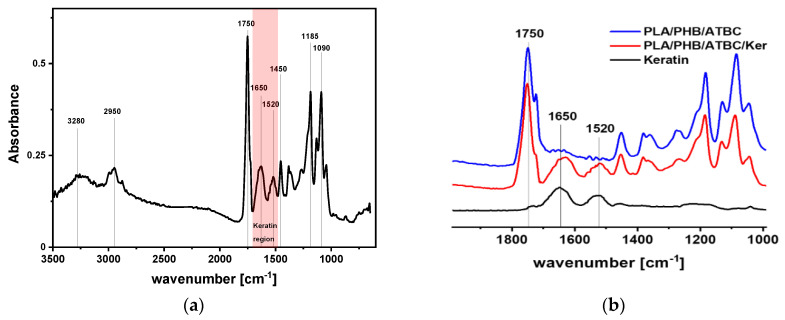
FTIR spectra of (**a**) PLA/PHB/ATBC/K-20 composite containing 20 wt% keratin and (**b**) comparison of this composite with keratin hydrolysate and PLA/PHB/ATBC matrix in MIR region; (**c**) neat polymer samples and keratin composites with 1, 3, 5, 10 and 20 wt% keratin filler content in PLA/PHB/ATBC matrix in NIR region and (**d**) optical microscopy picture at 300× magnification of PLA/PHB/ATBC/K-20 composite containing 20 wt% keratin.

**Figure 2 polymers-13-02693-f002:**
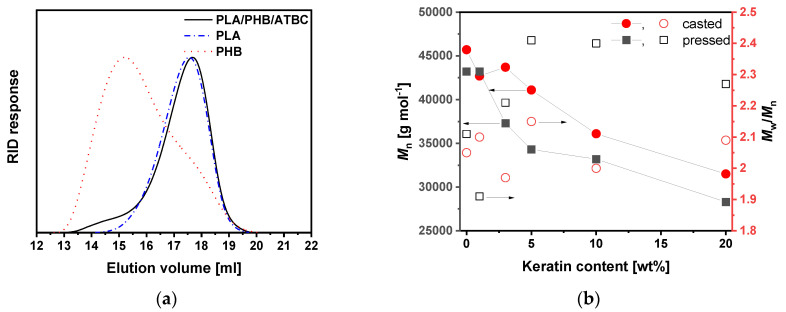
(**a**) GPC traces of PLA/PHB/ATBC blend and related polymers; (**b**) evolution of number average molar masses (*M*_n_) and dispersity (*Ð*) for casted and compression-molded samples of PLA/PHB/ATBC blend and its keratin composites with different keratin contents varying from 1 to 20 wt%.

**Figure 3 polymers-13-02693-f003:**
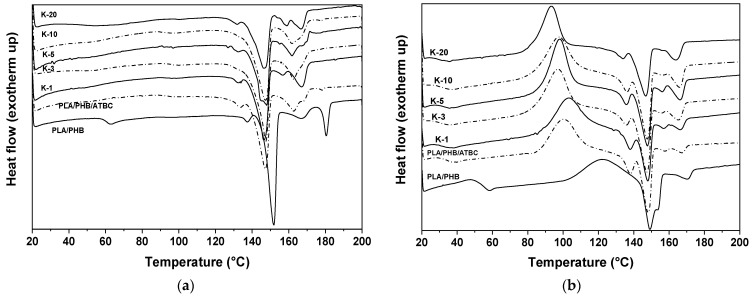
DSC (**a**) first and (**b**) second heating records of PLA/PHB and PLA/PHB/ATBC neat blends and their keratin composites with different keratin contents varying from 1 to 20 wt%.

**Figure 4 polymers-13-02693-f004:**
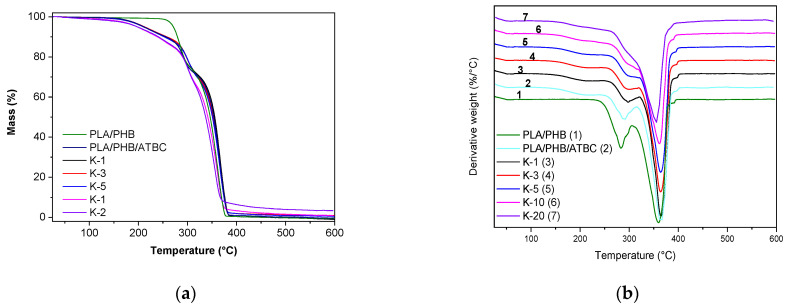
TGA (**a**) and DTG (**b**) records for neat PLA/PHB and PLA/PHB/ATBC blends and their keratin composites with different keratin contents varying from 1 to 20 wt%.

**Figure 5 polymers-13-02693-f005:**
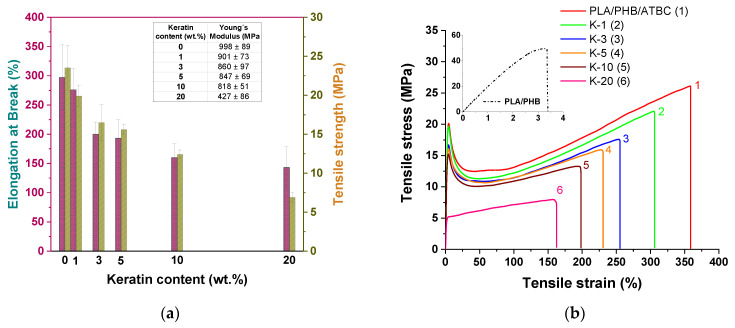
(**a**) Values of tensile properties (*ε*_B_ and *σ*_M_) and Young’s modulus (inserted table) and (**b**) stress–strain curves for PLA/PHB/ATBC blend and its keratin composites with different keratin contents varying from 1 to 20 wt%. The inset of Figure 5b shows the stress–strain curve for PLA/PHB blend.

**Figure 6 polymers-13-02693-f006:**
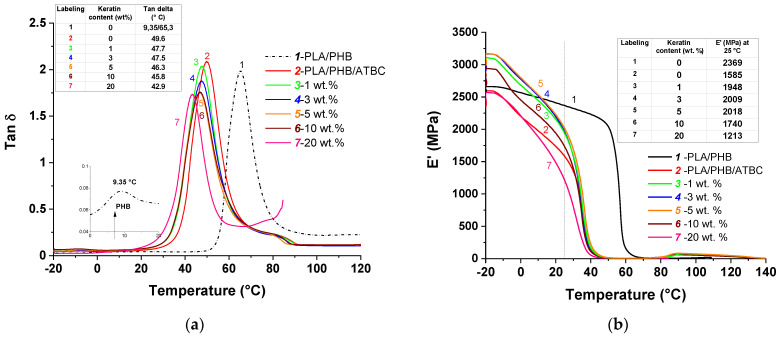
Evolution of loss factor tan δ (**a**) and dynamic modulus *E’* (**b**) with temperature for PLA/PHB/ATBC blend and its keratin composites with different keratin contents varying from 1 to 20 wt%.

**Figure 7 polymers-13-02693-f007:**
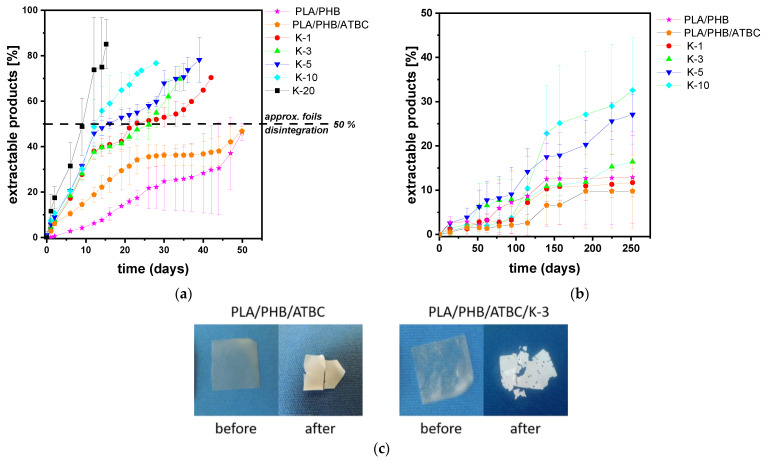
Extractable products during hydrolysis of PLA/PHB/ATBC blend and its keratin composites with different keratin contents varying from 1 to 20 wt%; (**a**) basic hydrolysis in 0.01 M NaOH; (**b**) hydrolysis in pure water; (**c**) representative pictures of PLA/PHB/ATBC blend and PLA/PHB/ATBC/K-3 composite before and after basic hydrolysis in 0.01 M NaOH.

**Figure 8 polymers-13-02693-f008:**
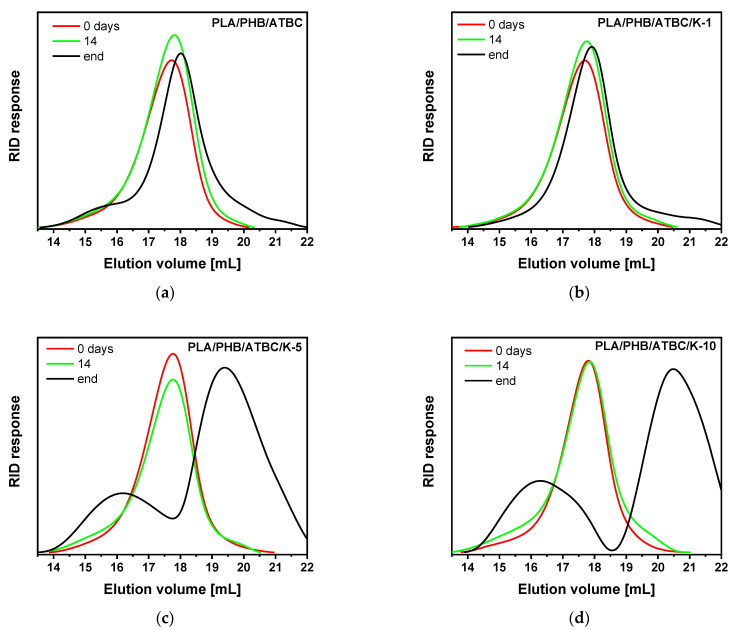
GPC traces of PLA/PHB/ATBC blend (**a**) and its keratin composites with 1 wt% (**b**), 5 wt% (**c**) and 10 wt% (**d**) before hydrolysis, after 14 days and at the end of hydrolysis in 0.01 M NaOH.

**Table 1 polymers-13-02693-t001:** Parameters obtained from DSC heating records of PLA/PHB and PLA/PHB/ATBC neat blends and their keratin composites with different keratin contents varying from 1 to 20 wt%.

Sample	*T*_g_ (°C)	*T*_cc_ (°C)	*T*_mPLA_ (°C)	*T*_mPHB_ (°C)	Δ*H*_m1_ ^a^ (J g^−1^)	Δ*H*_m2_ ^b^ (J g^−1^)	*χ*_c_ ^c^ (%)
PLA/PHB	54.1	123.8	152.6	173.3	47.69	44.99	47.20
PLA/PHB/ATBC	35.7	94.9	147.9	163.3	47.66	42.01	47.17
PLA/PHB/ATBC/K-1	36.8	98.3	147.6	165.6	37.59	36.64	36.26
PLA/PHB/ATBC/K-3	28.5	96.9	147.4	162.6	45.11	45.95	44.65
PLA/PHB/ATBC/K-5	31.7	98.2	147.3	163.3	33.74	34.13	33.39
PLA/PHB/ATBC/K-10	31.3	96.9	147.2	161.6	38.35	40.63	37.96
PLA/PHB/ATBC/K-20	30.1	93.4	147.1	166.3	32.56	35.88	32.23

^a^ Overall heat of melting from 1st heating run; ^b^ overall heat of melting from 2nd heating run; ^c^ calculated from enthalpies obtained in 1st heating run.

**Table 2 polymers-13-02693-t002:** Parameters obtained from TGA and DTG records for neat PLA/PHB and PLA/PHB/ATBC blends and their keratin composites with different keratin contents varying from 1 to 20 wt%.

Sample	*T*_onset_ (°C)	*T*_PHB_ (°C)	*T*_PLA_ (°C)	*R*_600 °C_ (%)
PLA	448.7	-	472.9	0
PHB	278.9	291.7	-	0
keratin	266.2	-	-	19.93
PLA/PHB	265.8	283.8	359.3	0
PLA/PHB/ATBC	276.3	287.2	364.9	0
PLA/PHB/ATBC/K-1	281.7	298.5	365.4	0
PLA/PHB/ATBC/K-3	284.3	298.9	364.6	0.93
PLA/PHB/ATBC/K-5	286.0	299.7	364.7	0.56
PLA/PHB/ATBC/K-10	285.2	300.7	361.6	3.35
PLA/PHB/ATBC/K-20	287.5	300.6	353.8	4.96

**Table 3 polymers-13-02693-t003:** Molecular characteristics and PLA/PHB ratio of PLA/PHB/ATBC blend and its keratin composites with different keratin contents varying from 1 to 20 wt% in various stages of hydrolysis in 0.01 M NaOH.

Sample	Before Hydrolysis		14 Days		End of Hydrolysis		
	*M*_n_, *M*_w_, (*Ð*) ^a^ g mol^−1^	PLA/ PHB ^b^	*M*_n_, *M*_w_, (*Ð*) ^a^ g mol^−1^	EH ^c^ %	*M*_n_, *M*_w_, (*Ð*) ^a^ g mol^−1^	PLA/ PHB ^b^	EH ^c^ %
PLA/PHB	51,200, 115,000 (2.25)	84/16	47,600, 126,700 (2.66)	8	10,200, 58,300 (5.69)	81/19	48
PLA/PHB/ATBC	45,400, 93,100 (2.05)	88/12	45,800, 109,600 (2.39)	22	19,500, 72,600 (3.73)	85/15	48
PLA/PHB/ATBC/K-1	45,400, 89,300 (1.97)	88/12	43,500, 92,400 (2.12)	40	6600, 48,300 (7.36)	84/16	70
PLA/PHB/ATBC/K-3	44,000, 89,400 (2.03)	87/13	39,200, 83,700 (2.13)	40	16,200, 63,700 (3.93) ^d^	82/18	70
PLA/PHB/ATBC/K-5	38,500, 82,300 (2.14)	88/12	38,800, 94,900 (2.44)	49	6100, 66,300 (10.8) ^d^	78/22	78
PLA/PHB/ATBC/K-10	37,800, 80,300 (2.12)	88/12	32,400, 89,000 (2.74)	55	5300, 42,600 (8.04) ^d^	75/25	77
PLA/PHB/ATBC/K-20	32,900, 76,900 (2.34)	88/12	29,700, 62,600 (2.11)	78	16,200, 50,900 (3.14) ^d^	67/33	86

^a^ Calculated based on PMMA standards; ^b^ based on ^1^H NMR integration peaks of PLA at 1.56 ppm and PHB at 1.26 ppm; ^c^ extent of hydrolysis based on extractable products analysis shown in Figure 5a; ^d^ significant bimodal molar mass distribution was observed.

**Table 4 polymers-13-02693-t004:** Concentration of bacteria microorganisms over the PLA/PHB/ATBC polymer blend and its keratin composites with different keratin contents varying from 1 to 20 wt%.

Sample	Concentration of Adherent Bacteria in Suspension (CFU/mL)
Soil suspension without polymers	3.00 × 10^6^
PLA/PHB/ATBC	5.10 × 10^9^
PLA/PHB/ATBC/K-1	8.51 × 10^8^
PLA/PHB/ATBC/K-5	7.50 × 10^8^
PLA/PHB/ATBC/K-10	5.00 × 10^8^
PLA/PHB/ATBC/K-20	4.00 × 10^8^

## Data Availability

Not applicable.

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
