# Peer review of "Properties and Degradation Performances of Biodegradable Poly(lactic acid)/Poly(3-hydroxybutyrate) Blends and Keratin Composites"

_polymers, 2021, doi:10.3390/polym13162693_

Round 1
Reviewer 1 Report
In the manuscript “Properties and Degradation Performances of Biodegradable Poly(lactic acid)/poly(3-hydroxybutyrate) Blends and Keratin Composites” the authors present a study where they analyze the effect of adding keratin to a biopolymers blend, the Keratin was used as a filler in plasticized blend. They found that incorporation of Keratin in plasticized PLA/PHB blend provides material with good thermal and mechanical properties and improved degradation In general, I observe that most of the results were presented in Katarína Mosnácková et al (2020) Int. J. Mol. Sci. 2020, 21, 9678; doi:10.3390/ijms21249678, for example:
- The formulations are the same.
- The tensile properties: these are presented in a graph, a histogram, and a table. the same information.
- The dynamic mechanical thermal analysis was already presented also in his previous article
- There is a difference whit the other article in FTIR analysis, However in Figure 1c) there are no much discussion.
I believe that this new article should focus on the new information, por example:
- DSC study
- TGA
- Extractable products during hydrolysis of……
- At the abstract is indicated that Hydrolytic degradation of matrices and composites was investigated by determination of extractable products amount, GPC, DSC and NMR???.
- Microbial test, etc…..
Additional remarks:
Page 4, line 164, microscope Leica DVM6 (indicate city and country fabrication)
Pag 4 line 170, Shimadzu UV-1650PC spectrometer (indicate city and country fabrication)
Pag 4, line 172 gel permeation chromatography (GPC), information of GPC?
Pag 14, Make sure the title of table 3 is on the same page.
use the same format for reference 2
Reviewer 2 Report
The manuscript presents very interested, detailed research on plasticized polylactic acid/poly(3-hydroxybutyrate) blend under loading in the range of 1-20 wt% of Keratin. The results have shown that the added Keratin promotes degradation of the matrices during the melt matrix, but the addition of the Keratin did not significantly affected the thermal behaviour of the PLA/PHB/ATBC blend.
This conclusion is the same as the one published at reference 2 (https://www.mdpi.com/1422-0067/21/24/9678,
Properties and Degradation of Novel Fully Biodegradable PLA/PHB Blends Filled with Keratin)
:
"The addition of keratin did not affect the extent of degradation of the PLA/PHB blend during melt blending."
Moreover, the conclusions in Abstracts are the same "he developed keratin-based composites possess properties comparable to commonly used thermoplastics applicable for example as packaging materials". Therefore there is a need to clearify the differences between this manuscript and the manuscript "https://www.mdpi.com/1422-0067/21/24/9678" more clearly.
The methodology is well prepared, all the results have sufficient data and there is no need for major or minor review.
Therefore the manuscript can be published as it is.
Round 2
Reviewer 1 Report
The revised manuscript has been improved.
Reviewer 2 Report
The manuscript is now more clear and can be publish as it is.